# NAQ: NONLINEARITY-AWARE QUANTIZATION

## ABSTRACT

Transformer-based large language models and vision transformers have achieved remarkable performance, but at a high energy cost. Nonlinearities (e.g., GELU, softmax) have regions where the magnitude of the gradient is small, which means that errors in pre-nonlinearity inputs result in small output error. We propose Nonlinearity-Aware Quantization (NAQ), which involves computing the FC layer outputs and attention scores at low precision, predicting the magnitude of the gradient of the nonlinearity, and recomputing the pre-nonlinearity if the gradient magnitude is large. With future hardware support, models with NAQ would avoid up to 62% of full precision pre-nonlinearity computation and it would achieve up to 29% reduction in energy consumption, with small effects on model performance.

## 1 INTRODUCTION

The transformer architecture (Vaswani et al., 2017) is the basis of large, pre-trained models that have achieved remarkable performance on language and vision modelling tasks, fuelling the recent excitement around generative AI. However, their growing energy consumption is a significant challenge, requiring the costly construction of new power plants to power this demand (Hiller, 2024). This increasing energy cost is hindering the availability and adoption of AI, such as on mobile platforms like smartphones and autonomous drones, requiring offloading of the most powerful models to cloud servers. A recent study shows model inference consumes 70% of AI infrastructure power capacity compared to 10% for experimentation and 20% for training (Wu et al., 2022). This motivates the need for solutions that reduce the inference energy consumption of existing, trained models.

Common to all transformer architectures are building blocks of linear layers followed by an activation function (e.g., GELU) and attention weight computation followed by softmax. We find that some of this pre-nonlinearity (pre-activation and pre-softmax) computation can be computed at low precision with little effect on model performance, because nonlinearities have regions where the magnitude of the gradient is small. Fig. 1 (left) shows gradient of various element-wise, nonlinear activation functions plotted against pre-activation input value, and it shows that there are regions where the

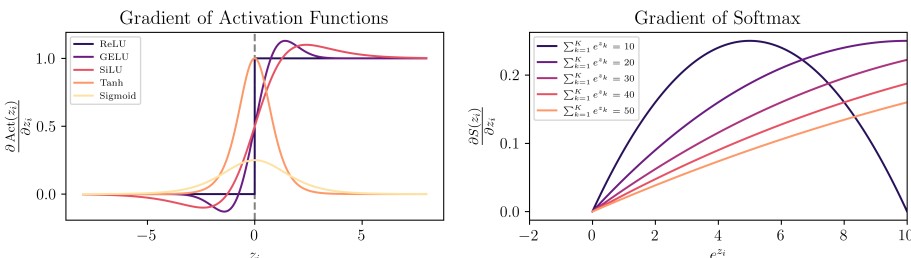

Figure 1: (left) Gradients of activation functions is plotted against the input. Note that when the input is negative, the gradients of activation functions tend to have a small magnitude. (right) Gradient of the softmax function is plotted against the input exponential $e^{z_i}$ at a selection of values for the sum of exponentials $\sum_{k=1}^{K} e^{z_k}$. Note that the gradient is negatively correlated to the sum of exponentials and is positively correlated to the input exponential.

gradient has small magnitude. Fig. 1 (right) plots the gradient of softmax against the exponential of a given attention score $e^{z_i}$ for various values for sum of exponentials $\sum_{k=1}^{K} e^{z_k}$ (shown in the legend). As the attention score exponential decreases or the sum of exponentials increases, the gradient magnitude decreases. When the gradient magnitude is small, errors in the pre-nonlinearity have little effect on the output of the nonlinearity, which presents an opportunity for less precise computation.

Since the gradients of nonlinearities are functions of the pre-nonlinearity values (as shown in Fig. 1), pre-computing the gradient to determine how to quantize weights and inputs is not an option. In other words, the dilemma is that we need to know the result of the pre-nonlinearity computation before we know how to quantize the inputs to the pre-nonlinearity computation. We break this dependency loop by performing low-precision computation to gather information about the gradient magnitude, and then recomputing where the gradient magnitude is large. Since low-precision computation uses much less energy than full precision, the energy savings outweigh the cost of recomputation. We call this technique NAQ, and it specifically targets pre-activation fully-connected (FC) layer and the pre-softmax attention score computation. While prior works have also proposed hardware solutions predicting when the input to ReLU will be negative to avoid computation that will end being set to 0 (Akhlaghi et al., 2018; Kim et al., 2021a), their assumptions break down when applied to other nonlinearities (§A). In contrast, our insights lead to a proposal called NAQ, which improves the energy efficiency for computation preceding all activation functions and softmax.

We make the following contributions:

- We observe that common nonlinearities in transformers (e.g., GELU and softmax) have regions of small gradient magnitude.

- We propose a technique called Nonlinearity-Aware Quantization (NAQ) that gathers information about the gradient with a low-precision pass before selectively recomputing pre-nonlinearity values where the gradient is predicted to be large.

- We apply NAQ on vision transformers (ViTs) and large language models (LLMs) quantized using GPTQ (Frantar et al., 2022) and PTQ4ViT (Yuan et al., 2022), resulting in avoiding up to 62% of full precision pre-nonlinearity computation and achieving up to 29% reduction in energy consumption, with small effects on model performance.

## 2 BACKGROUND AND MOTIVATION

In this section, we describe the components of the transformer architecture targeted by this work, analyze the gradients of nonlinearities, and profile transformers to provide the relevant context and motivation for our approach.

### 2.1 RELATED WORK

Transformer Quantization for inference has been heavily studied (Gholami et al., 2022) with quantization-aware training (QAT) (Bai et al., 2020; Zhang et al., 2020a; Kim et al., 2021b; Shen et al., 2020; Zafrir et al., 2019) and post-training quantization (PTQ) (Xiao et al., 2023; Frantar et al., 2022; Yuan et al., 2022; Kim et al., 2023; Yao et al., 2022; Lin et al., 2024). Since QAT requires re-training, we opt to focus on a PTQ-base approach for this work. In general, most recent PTQ work has rightly aimed to decrease model size and memory bandwidth requirements. While those are important goals, in this paper, we tackle a less explored goal of decreasing energy used in computation.

(Kim et al., 2023) has similarities to this paper in that it represents hard to quantize (sensitive) values in full precision, but our work differs in a few key ways. SqueezeLLM uses the full model Hessian to estimate sensitivity instead of nonlinearity gradient in this work. This means that our work does not require any of backpropagation steps that SqueezeLLM does. Additionally, SqueezeLLM finds sensitive values offline, whereas in this work, large gradient magnitude values are determined dynamically after low-precision computation.

Some works propose ReLU-based early negative termination (Akhlaghi et al., 2018; Kim et al., 2021a), which are hardware approaches that terminate computation early if the output is predicted

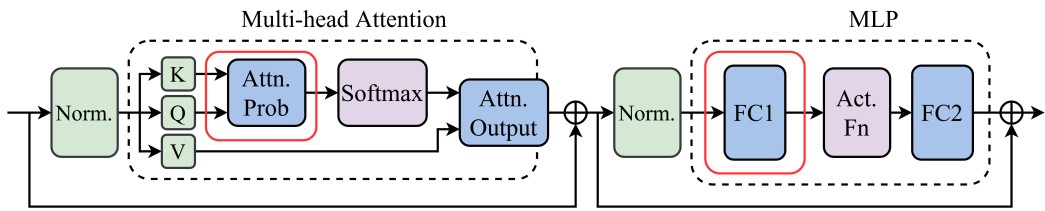

Figure 2: NAQ reduces energy consumption in the pre-softmax attention probability computation and in the pre-activation FC layer.

to be negative and consequently set to zero by ReLU. This approach does not extend to non-RELU activation functions, nor softmax. See App. A for details.

## 2.2 THE TRANSFORMER ARCHITECTURE

Fig. 2 illustrates the modern transformer architecture, which is based on (Vaswani et al., 2017), and highlights in red the pre-nonlinearity computations that NAQ targets. In the attention block, matrix multiplication between query and key tensors yields attention scores, which are converted into attention weights by applying softmax, and multiplying these weights by the value tensor produces the output of the attention block. While there is some variation in implementation of attention blocks (Liu et al., 2021; Yu et al., 2022; Beltagy et al., 2020; Kitaev et al., 2020), attention blocks generally follow the pattern of linear transformation followed by softmax to produce attention weights. In the MLP block, there are typically two FC layers with an activation function layer in between. The common activation functions for transformers are GELU (Devlin et al., 2018; Workshop et al., 2022; Touvron et al., 2021; Radford et al., 2018; Liu et al., 2021; Dosovitskiy et al., 2021), SwiGLU (a linear transformation of SiLU) (Touvron et al., 2023), and ReLU (Zhang et al., 2022). NAQ supports many activations functions, even those not found in transformers (see App. B).

## 2.3 GRADIENT OF NONLINEARITIES

When the magnitude of the gradient of the output of a nonlinearity is small for a given value in a tensor, then that value would be a prime candidate for lower precision representation and computation, since the error from low precision computation would result in less output error. In other words, quantization error would be mitigated by the low gradient magnitude. This is similar to the idea of "sensitive values" in SqueezeLLM (Kim et al., 2023), where values that are estimated to be likely to impact the loss of the network through a Hessian heuristic are represented in full-precision. In our case, we consider "sensitive values" in terms of their direct effect on the output of the nonlinearity by estimating the gradient. In this subsection, we analyze nonlinearities to determine when the magnitude of their gradient is small with respect to elements of the pre-activation tensor and inputs to linear transformations that produce the pre-activation tensor.

### 2.3.1 FC LAYER AND ACTIVATION FUNCTION

Consider the innermost loop of the FC layer and activation function, for an output value $y_i$:

$$y_i = \text{Act}(w \cdot x + b) \tag{1}$$

where $w$ is the weight vector, $b$ is the bias term, $x$ is the input vector, and $\text{Act}$ is the activation function. For convenience, define a pre-activation value as $z_i = w \cdot x + b$, so $y = \text{Act}(z_i)$.

The gradient of $\text{Act}(z_i)$ is depends on $z_i$ and this function $f(z_i)$ is plotted for several activation functions in Fig. 1. Thus, the gradient of the output $y_i$ with respect to individual elements of $w$ and $x$ are functions of $z_i$:

$$\frac{\partial y}{\partial x_i} = \frac{\partial \text{Act}}{\partial z_i} \frac{\partial z_i}{\partial x_i} = \frac{\partial \text{Act}}{\partial z_i} w_i = f(z_i)w_i \tag{2}$$

$$\frac{\partial y}{\partial w_i} = \frac{\partial \text{Act}}{\partial z_i} \frac{\partial z_i}{\partial w_i} = \frac{\partial \text{Act}}{\partial z_i} x_i = f(z_i)x_i \tag{3}$$

As the gradient of the activation function depends on the pre-activation ($z$), it is not possible to perfectly determine the gradient of the output with respect to the input without computing the pre-activation. In other words, directly computing the gradient is not an option for determining how to quantize individual weight and input elements. NAQ (§3) tackles this problem through low-precision computation of the FC layer and recomputation where gradient magnitude is predicted to be large.

### 2.3.2 ATTENTION WEIGHT

The gradient of attention weights (output of softmax) with respect to attention scores (pre-softmax values) requires completing the pre-softmax computation, not dissimilar to the dependency loop for the gradient of activation functions. Since the calculation of attention scores varies from model to model, we consider a generic linear transformation. Consider the computation of an attention weight slice (or vector) of length $K$, as a one-dimensional softmax is typically performed along the last dimension of the attention score tensor. The attention weight is given by

$$w_a = S(\text{F}_s(x, \theta)) \tag{4}$$

where $w_a$ is the attention weight slice, $S$ is the softmax function, $\text{F}_s$ is the linear transformation on the inputs, $x$, using parameters, $\theta$, which can include layer weights (to map $x$ to keys and queries) and masks. For convenience, define the attention score $z = \text{F}_s(X)$, so $w_a = S(z)$.

The $i^{\text{th}}$ element of the attention weight slice is given by

$$S(z_i) = \frac{e^{z_i}}{\sum_{k=1}^{K} e^{z_k}}, \tag{5}$$

where $K$ is the size of slice $z$. The partial gradient of the $i^{\text{th}}$ attention weight element with respect to the $j^{\text{th}}$ attention score is

$$\frac{\partial S(z_i)}{\partial z_j} = S(z_i)(1\{i = j\} - S(z_j)) = \frac{e^{z_i}}{\sum_{k=1}^{K} e^{z_k}} \left( 1\{i = j\} - \frac{e^{z_j}}{\sum_{k=1}^{K} e^{z_k}} \right) \tag{6}$$

where $1\{i = j\}$ is 1 if $i = j$ and 0 otherwise (Kuribel, 2021).

Eqn. 6 shows that computing the gradient of the softmax requires computing the softmax itself. We can learn when magnitude of the gradient would be larger or smaller by analyzing the numerators and denominators in the equation, with the eventual goal of computing these heuristics with energy-efficient low-precision computation. The exponential of the input $e^{z_i}$ is in the denominator and the sum of exponentials $\sum_{k=1}^{K} e^{z_k}$ is in the denominator, so we see the following relationships:

$$\frac{\partial S(z_i)}{\partial z_j} \propto e^{z_i} \quad \text{and} \quad \frac{\partial S(z_i)}{\partial z_j} \propto \frac{1}{\sum_{k=1}^{K} e^{z_k}} \tag{7}$$

Fig. 1 illustrates the positive correlation between $\frac{\partial S(z_i)}{\partial z_j}$ and $e^{z_i}$ and the negative correlation between $\frac{\partial S(z_i)}{\partial z_j}$ and $\sum_{k=1}^{K} e^{z_k}$.

## 2.4 QUANTIZATION

Quantization reduces the number of bits required to represent a given value, by applying a mapping function $f$ to map the high precision value, $x_{\text{HP}}$, to the range [-1,1], multiplying by the maximum quantized representation for a given $N$ bits and rounding the result, and dequantization works in reverse:

$$x_{\text{INT}} = \text{round}\left(\left(2^{N-1} - 1\right) f(x_{\text{HP}})\right) \tag{8}$$

$$x_{\text{HP, dequantized}} = f^{-1}\left(\frac{x_{\text{INT}}}{2^{N-1} - 1}\right) \tag{9}$$

The maximum quantized representation is $2^{N-1} - 1$ because one bit is used as sign bit. For simplicity, we use a symmetric quantization function $f(x) = \frac{x}{M}$, where $M$ is the maximum representable value. However, the quantization function can become more complex with the addition of zero-point (Jacob et al., 2018), group-wise quantization (Shen et al., 2020), and non-uniform quantization (Kim et al., 2023).

Quantization introduces quantization error through the rounding process, but computation with quantized values uses much less energy. Energy consumption for multiplications scales quadratically (a multiplication with half the bits will take one quarter the energy), while energy consumption for additions, scales linearly (Horowitz, 2014). This relationship is illustrated in (Horowitz, 2014), where the author lists the energy consumption of computing operations at different bitwidths. Using this relationship, we estimate that at 45nm technology, an INT4 multiply-add costs 0.065pJ while a 16-bit floating point multiply-add costs 1.5pJ and INT8 multiply-add costs 0.23pJ, so an INT4 multiply-add uses $23.1\times$ less energy than FP16 and $3.5\times$ less than INT8. The relative difference in energy consumption holds at other process nodes (Stillmaker & Baas, 2017). This highlights the tradeoff between energy consumption and model performance, where we aim to limit the model performance degradation as energy consumption is quadratically decreased.

## 2.5 PROPORTION OF ENERGY CONSUMPTION BY PRE-NONLINEARITY COMPUTATION

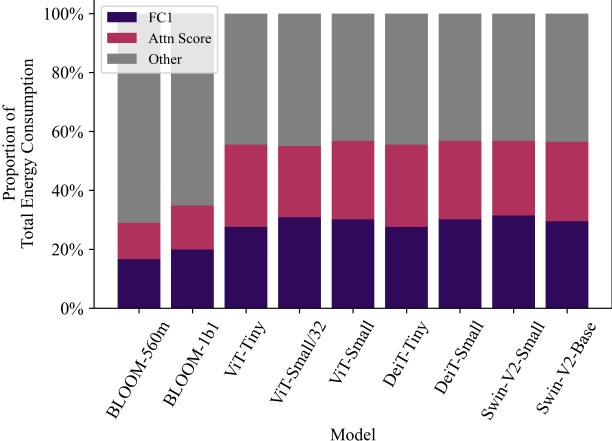

Figure 3: The pre-activation FC layer and the pre-softmax attention score computation combined account for 29%-57% of the energy consumed by computation.

In our baselines (§4.1), GPTQ performs computation in FP16 (after dequantizing weights from INT4) (Frantar et al., 2022), while PTQ4ViT performs computation in INT8 (Yuan et al., 2022). We measured multiply-adds using the `torchprofile`[1] tool. As all multiply-adds are either FP16 or

---

[1]https://github.com/zhijian-liu/torchprofile

INT8, we assign the proportion of computation energy consumed as the proportion of total multiply-adds required for inference. Fig. 3 shows that the pre-activation FC layer makes up 16%-31% of the total computation energy consumption of the transformer model and the pre-softmax attention score computation 12%-28%. Combined, the pre-nonlinearity computation consumes 29%-57% of the total computation energy. By targeting this energy intensive computation, we significantly reduce the overall amount of energy consumed by the model.

## 3 NONLINEARITY-AWARE QUANTIZATION

We propose Nonlinearity-Aware Quantization (NAQ) to exploit the small gradient magnitude regions in nonlinearities (§2.3) in order to reduce the energy consumption of pre-nonlinearity computation (§2.5). In §2.3, we described a dependency loop where we need the nonlinearity gradients to selectively quantize the inputs, but we need to compute the pre-nonlinearity values from these inputs to determine the nonlinearity gradients. NAQ predicts when the gradient is large via low-precision computation and then uses the prediction to recompute at full precision when the gradient magnitude is large, thus breaking the dependency loop.

NAQ addresses energy efficiency because computing at low-precision uses much less energy than full precision (§2.4) and we only recompute a fraction of the total computation at full precision. However, recomputation adds latency as the critical path includes the original full precision computation and additional low-precision computation. For this reason, NAQ is not ideal for latency-sensitive applications, but NAQ is suited to energy-constrained applications, such as data centers that would like to lower energy costs and mobile platforms that are limited by battery life.

Quantization and dequantization are performed according to Eqn. 8 and Eqn. 9, respectively. Fixed point low-bitwidth computation is performed in the dot product and addition in the FC layer (Fig. 4 (left)), and in attention score computation (Fig. 4 (right)) if the parameters are quantized by model compression. In §4.1, we describe building on top of quantization-based model compression works.

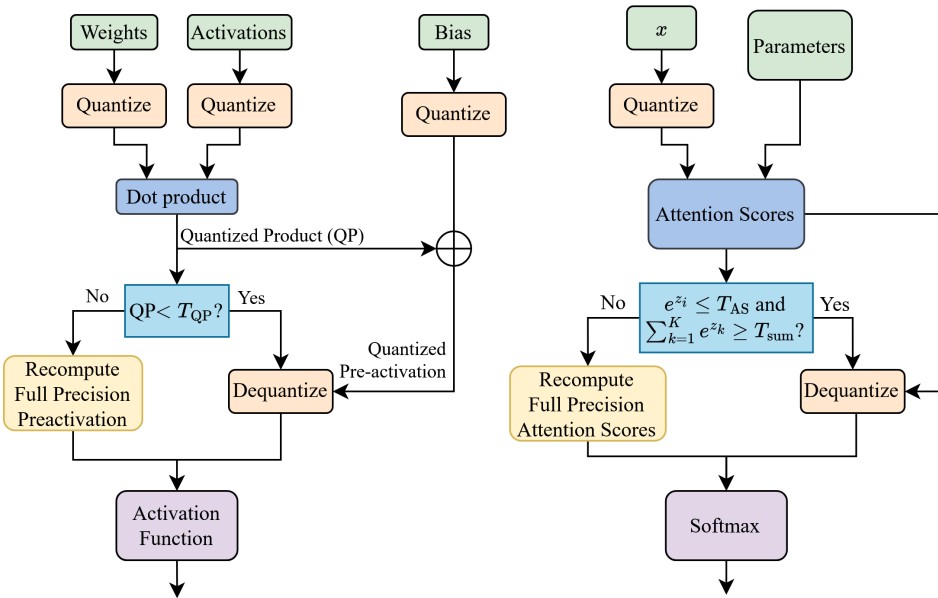

Figure 4: Flowchart demonstrating applying the NAQ technique to the pre-activation FC layer (left) and the pre-softmax attention score computation (right). The pre-nonlinearity computation is computed on quantized values to learn nonlinearity gradient information at lower energy consumption, then the gradient information informs how to selectively recompute where the gradient magnitude is predicted to be large.

## 3.1 FULLY-CONNECTED LAYER

The flowchart in Fig. 4 (left) shows how NAQ is applied to a fully-connected layer computation that produces the pre-activation tensor. First, the weights, activations, and biases fetched (green) and quantized (beige). Second, the quantized weights and activations are multiplied (dark blue) producing the quantized product (QP), then the quantized bias is added to produce the quantized pre-activation (QPA). Then, the QP is compared against the QP threshold ($T_{QP}$) on an element-wise basis (light blue):

- If $QP \leq T_{QP}$, then the QPA is dequantized (beige) and passed into the activation function.

- If $QP > T_{QP}$, then the QPA is discarded and the full-precision pre-activation (yellow) is computed and passed into the activation function.

We found that QP is a good predictor for the gradient. Increasing $T_{QP}$ reduces the amount of recomputation at the cost of allowing more quantization error.

## 3.2 ATTENTION SCORE

The flowchart in Fig. 4 (right) shows how NAQ quantizes the attention score computation that produces the pre-softmax tensor. First, attention score is computed on quantized keys and queries. Second, the exponential of attention scores ($e^{z_i}$) and the sum of exponentials along the last dimension of the tensor ($\sum_{k=1}^{K} e^{z_k}$) are compared to thresholds on an element-wise basis:

- If $e^{z_i} \leq T_{AS}$ and $\sum_{k=1}^{K} e^{z_k} \geq T_{sum}$, then the quantized attention score is dequantized and passed into the softmax.

- If not, then the quantized attention is discarded and the full-precision pre-activation is computed and passed into the activation function.

Both $e^{z_i}$ and $\sum_{k=1}^{K} e^{z_k}$ affect the gradient magnitude (see §2.3.2 and Fig. 2.3), so use both as heuristics to predict how large the gradient magnitude will be.

## 3.3 QUANTIZATION THRESHOLDS

NAQ uses quantization thresholds on the quantized product for the pre-activation fully-connected layer and on $e^{z_i}$ and $\sum_{k=1}^{K} e^{z_k}$ for the pre-softmax attention score computation. This is because NAQ computes quantized pre-nonlinearity tensors, which would yield inaccurate gradient values. We can empirically show a tradeoff in model performance and energy saved by changing the quantization thresholds. If there is more full-precision recomputation, then more energy will be used, but the model performance would also improve (§4).

## 3.4 HARDWARE IMPLEMENTATION

A limitation of NAQ is that there is not current hardware support to achieve energy efficiency gains. However, commodity hardware and computer architecture research is not far from being able to support NAQ. NVIDIA GPUs have support INT4 computation since 2019, with throughput around double INT8 (Dave Salvator & Emmart, 2019). Assuming the same power demand for INT4 compared to INT8, this would mean the energy per computation scales linearly, as opposed to quadratically like we believe is possible with specialized hardware. While commodity GPUs do not directly support the arbitrary sparse computation during full-precision recomputation, it is possible to repack the sparse computation into the 2:4 structured sparsity pattern (Choquette, 2023), though this would come with some data movement overheads in repacking. Recent computer architecture works (Zhang et al., 2020b; Srivastava et al., 2020) have shown one to two orders of magnitude improvements in throughput and energy efficiency compared to GPUs. We hope this study motivates further research and development of better support for sparsity.

# 4 METHODOLOGY AND EVALUATION

We study the potential of NAQ by implementing two PyTorch modules, one that replaces the first fully-connected layer and the subsequent activation function in the MLP block, and another that replaces the attention score computation and softmax in the Attention block. While these modules are not designed to show energy savings on commodity GPUs, they track the amount of full-precision recomputation, so that we can estimate the energy savings should specialized hardware be built. Quantization is performed as described in §2.4, with 4 bits and $M = 4$ for inputs and $M$ being the maximum absolute value weight along the last dimension of the weight tensor. Energy consumption estimated from counting full precision and 4-bit multiply-adds and using the energy estimation in §2.4

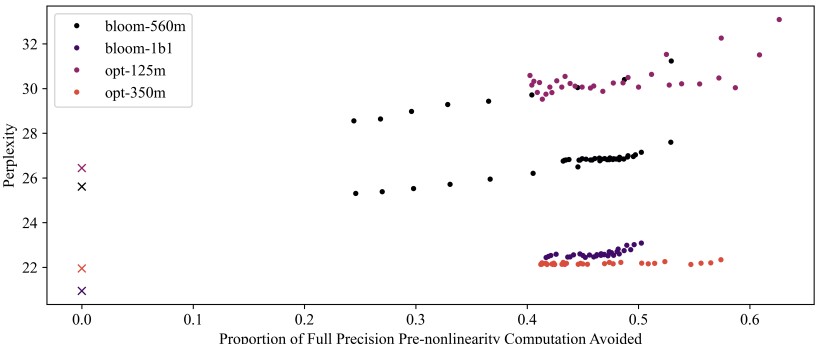

Figure 5: Perplexity of BLOOM and OPT GPTQ models when NAQ is applied to pre-nonlinearity computation. The plot shows the trade-off between perplexity and the proportion of full-precision computation that NAQ avoids computing as we sweep NAQ quantization thresholds. $\times$ represents the baseline perplexity of the GPTQ model without NAQ.

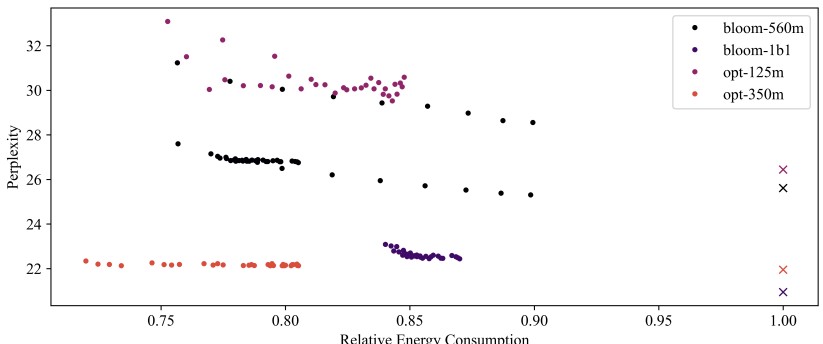

Figure 6: Perplexity of BLOOM and OPT GPTQ models when NAQ is applied to pre-nonlinearity computation. The plot shows the trade-off between perplexity and the relative energy consumption (lower is better) as we sweep NAQ quantization thresholds. $\times$ represents the baseline perplexity of the GPTQ model without NAQ.

## 4.1 BASELINES

Model compression is important for reducing the memory footprint and memory bandwidth usage of models, and it is synergistic with energy efficiency in terms of saving costs in data center hardware and energy use, as well as helping in deployment to mobile platforms. Thus, we evaluate all models building on prior quantization-based model compression works, PTQ4ViT (Yuan et al., 2022) and

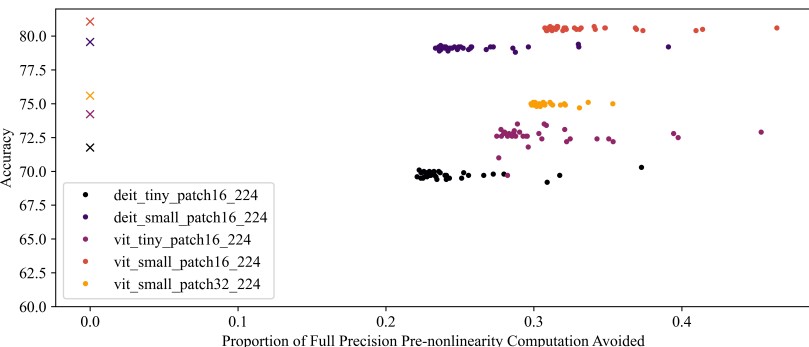

Figure 7: ImageNet (Deng et al., 2009) top-1 accuracy plotted against the proportion of full-precision computation that NAQ avoids computing for the PTQ4ViT ViT and DeiT models when NAQ is applied to its pre-nonlinearity computation with a sweep of quantization thresholds. × represents the baseline accuracies of the pre-trained model without NAQ.

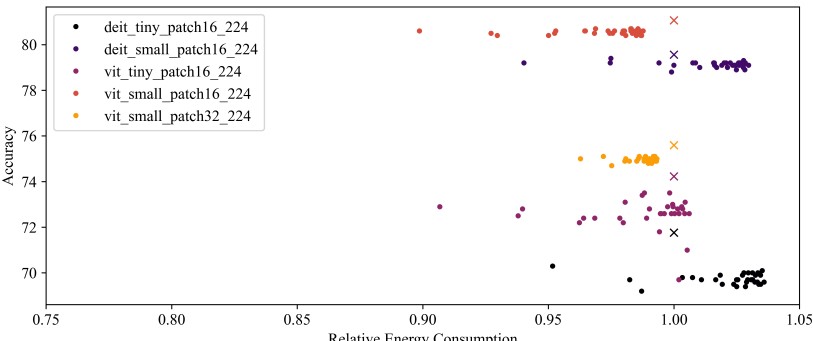

Figure 8: ImageNet (Deng et al., 2009) top-1 accuracy plotted against relative energy consumption of the PTQ4ViT ViT and DeiT models when NAQ is applied to its pre-nonlinearity computation with a sweep of quantization thresholds. × represents the baseline accuracies of the pre-trained model without NAQ.

GPTQ (Frantar et al., 2022). For PTQ4ViT models, we use the W8A8 baseline, so INT8 computation as the baseline. The dataset is ImageNet-1K, performance measured as top-1 accuracy (higher is better). We evaluate ViT (Dosovitskiy et al., 2020) and DeiT (Touvron et al., 2021) models. For GPTQ, we use the 4-bit weight baseline, but computation is done in FP16 as the baseline. The dataset is c4 (Raffel et al., 2020), evaluating perplexity (lower is better). We evaluate on BLOOM (Workshop et al., 2022) and OPT (Zhang et al., 2022).

## 4.2 MODEL PERFORMANCE AND ENERGY SAVINGS TRADE-OFF

In our evaluation, we show how much full precision pre-nonlinearity computation (FP16 for GPTQ and INT8 for PTQ4ViT) can be avoided without significant degradation in model performance. Then, we apply our model of how computation energy consumption is reduced by quantization (§2.4) to determine how much energy the NAQ models consume relative to the baseline, and plot this against model performance. We account for the overhead of computing low precision values that end up not being used because they are recomputed.

Results for applying NAQ on GPTQ models are shown in Fig. 5 and Fig. 6. NAQ can avoid 24%-62% of the full precision pre-nonlinearity computation with relatively small effects on perplexity. This

translates to reducing the relative energy consumption to as low as 0.71, saving 29% of the total energy consumption of the model. We speculate that the zero-gradient of the ReLU helps OPT-350M achieve the best perplexity at the lowest relative energy consumption.

Results for PTQ4ViT models with NAQ are shown in Fig. 7 and Fig. 8. While over 40% of full precision pre-nonlinearity computation can be avoided with small degradations in accuracy, NAQ struggles to reduce energy consumption. At most, it achieves a 10% energy savings. The reason this is the case is that full-precision computation for PTQ4ViT is INT8, which uses just $3.5\times$ more energy than INT4 computation. The additional energy for INT4 computation that is discarded and recomputed means the energy efficiency gains are more modest.

More detailed results with the quantization thresholds and more detailed breakdown of computation avoided can be found in App. C.

## 5  CONCLUSION

In this work, we propose NAQ, which exploits the small gradient magnitude regions of nonlinearities in the transformer architecture. Where the gradient magnitude is small, quantization error can be tolerated. Under NAQ, all pre-nonlinearity computation is performed at low bit-width, which provides information about the which pre-nonlinearity elements correspond to high gradient magnitude, so those can be re-computed at full precision. We evaluate NAQ by building on top of prior model compression work and we find that with hardware support, models with NAQ would avoid up to 62% of full precision pre-nonlinearity computation and it would achieve up to 29% reduction in energy consumption, with small effects on model performance. We hope our work stimulates further research into the interplay between the choice of activation function and accelerated approximate computing.

## 6  REPRODUCIBILITY STATEMENT

All of the results can be reproduced by using the code available at https://github.com/nonlinearity-aware-quantization/naq.

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

## A  RELU-BASED APPROACHES

In this section, we discuss in more depth the problems with applying ReLU-based prior approaches to transformers, as well quantitative results of trying to do so.

We consider SnaPEA (Akhlaghi et al., 2018), a representative early termination technique that exploits characteristics of ReLU in CNNs with the following two observations: (1) the ReLU function output zero for any negative input, so if one can predict which pre-activations will be negative, then one can skip computing the value of that negative pre-activation; and (2) many CNNs have repeating sequences of convolution followed by ReLU, which means that the input to most convolution layers are strictly non-negative. SnaPEA (Akhlaghi et al., 2018) orders weights from most positive to most negative, then computes and accumulates partial products serially until the partial product becomes negative. From observation (2), the partial products must decrease monotonically as they are computed, and from observation (1), it outputs zero once the partial product becomes negative.

The two assumptions underlying ReLU-based Early Negative Termination both fall apart in transformers. Observation (1) fails for non-RELU activation functions, because it is not enough to make a binary (output = 0 or something else) prediction when negative pre-activation values map to non-zero negative post-activation values. We swept a range of predicted single values and none achieve the performance of NAQ, despite the benefit of an unrealistic oracle predictor. Even if the transformer used ReLU like OPT (Zhang et al., 2022) does, an early termination method would run into problems with observation (2) not being true. As shown in Fig. 2, the inputs into the pre-activation computation are not the output of ReLU and thus are not strictly non-negative, violating that assumption underlying the early termination methods.

## B  TANH AND SIGMOID ACTIVATION FUNCTIONS

Tanh and sigmoid activation functions are not commonly used in transformer models, but they play an important role in the history of neural networks. Indeed, the Universal Approximation Theorem underlying all neural networks is proven using the sigmoid activation function (Hornik et al., 1989). Tanh and sigmoid are different from the other activations because their gradient $\frac{\partial \operatorname{Act}(z_i)}{\partial z_i}$ is symmetric around the y-axis with low gradient for large magnitude (absolute value) inputs. This contrasts to the other activations that are asymmetric, as shown in Fig. 1. Having low gradients for large magnitude inputs (regardless of sign) makes tanh and sigmoid helpful in combatting the problem of outliers in transformers (Xiao et al., 2023). Outliers are difficult to quantize as they can require more bits to accurately represent the dynamic range, but the low gradient for these outliers means that larger quantization error in representing these values would have a relatively small impact on the output.

Implementing NAQ on a neural network with tanh or sigmoid would require the quantization threshold $T_{textQP}$ to be applied to the absolute value of QP (§3.1) to predict the activation function gradient. This small change should suffice in enabling NAQ support for tanh and sigmoid. Further exploration of this would be warranted if tanh and sigmoid regain popularity.

## C  PARAMETERS AND PROPORTION OF COMPUTATION AVOIDED

We show the break down of the data shown in §4.2 by quantization thresholds $T_{\text{sum}}$, $T_{\text{AS}}$, $T_{\text{QP}}$. The data shows the proportion of full precision computation avoided due to the thresholds $P_{\text{QP}}$, $P_{\text{AS}}$, and $P_{\text{sum}}$, as well as the proportion avoided in the pre-softmax computation $P_{\text{sm}}$, and total avoided computation $P_{\text{NAQ}}$.

| $T_{\text{sum}}$ | $T_{\text{AS}}$ | $T_{\text{QP}}$ | $P_{\text{QP}}$ | $P_{\text{AS}}$ | $P_{\text{sum}}$ | $P_{\text{sm}}$ | $P_{\text{NAQ}}$ | Perplexity |
|---|---|---|---|---|---|---|---|---|
| 12800 | -1 | -0.00002 | 0.549 | 0.624 | 0.658 | 0.316 | 0.449 | 26.86 |
| 12800 | -2 | -0.00002 | 0.548 | 0.610 | 0.658 | 0.312 | 0.447 | 26.8 |
| 12800 | -3 | -0.00002 | 0.549 | 0.601 | 0.658 | 0.309 | 0.446 | 26.8 |
| 12800 | 0 | -0.00002 | 0.549 | 0.642 | 0.658 | 0.324 | 0.453 | 26.84 |
| 1600 | -1 | -0.00002 | 0.548 | 0.623 | 0.750 | 0.381 | 0.477 | 26.83 |
| 1600 | -2 | -0.00002 | 0.548 | 0.610 | 0.750 | 0.373 | 0.474 | 26.82 |
| 1600 | -3 | -0.00002 | 0.549 | 0.600 | 0.750 | 0.369 | 0.472 | 26.82 |
| 1600 | 0 | -0.00002 | 0.549 | 0.641 | 0.750 | 0.395 | 0.483 | 26.93 |
| 200 | -1 | -0.00002 | 0.548 | 0.623 | 0.800 | 0.424 | 0.495 | 26.96 |
| 200 | -3 | -0.00001 | 0.622 | 0.600 | 0.800 | 0.404 | 0.529 | 27.6 |
| 200 | -3 | -0.00001 | 0.623 | 0.600 | 0.801 | 0.404 | 0.529 | 31.24 |
| 200 | -3 | -0.00002 | 0.548 | 0.600 | 0.800 | 0.404 | 0.487 | 26.85 |
| 200 | -3 | -0.00002 | 0.549 | 0.600 | 0.801 | 0.404 | 0.487 | 30.41 |
| 200 | -3 | -0.00003 | 0.477 | 0.600 | 0.800 | 0.404 | 0.445 | 26.5 |
| 200 | -3 | -0.00003 | 0.476 | 0.600 | 0.801 | 0.404 | 0.445 | 30.05 |
| 200 | -3 | -0.00004 | 0.406 | 0.600 | 0.800 | 0.404 | 0.405 | 26.21 |
| 200 | -3 | -0.00004 | 0.404 | 0.599 | 0.801 | 0.404 | 0.404 | 29.72 |
| 200 | -3 | -0.00005 | 0.339 | 0.600 | 0.800 | 0.403 | 0.367 | 25.95 |
| 200 | -3 | -0.00005 | 0.337 | 0.599 | 0.801 | 0.404 | 0.365 | 29.44 |
| 200 | -3 | -0.00006 | 0.277 | 0.600 | 0.800 | 0.403 | 0.331 | 25.72 |
| 200 | -3 | -0.00006 | 0.273 | 0.599 | 0.801 | 0.404 | 0.329 | 29.29 |
| 200 | -3 | -0.00007 | 0.220 | 0.600 | 0.800 | 0.403 | 0.298 | 25.53 |
| 200 | -3 | -0.00007 | 0.216 | 0.600 | 0.800 | 0.403 | 0.296 | 28.98 |
| 200 | -3 | -0.00008 | 0.168 | 0.600 | 0.800 | 0.403 | 0.268 | 28.64 |
| 200 | -3 | -0.00008 | 0.171 | 0.600 | 0.799 | 0.403 | 0.270 | 25.39 |
| 200 | -3 | -0.00009 | 0.126 | 0.600 | 0.800 | 0.403 | 0.244 | 28.56 |
| 200 | -3 | -0.00009 | 0.129 | 0.600 | 0.799 | 0.403 | 0.246 | 25.31 |
| 200 | 0 | -0.00002 | 0.548 | 0.641 | 0.800 | 0.441 | 0.502 | 27.15 |
| 25600 | -1 | -0.00002 | 0.549 | 0.624 | 0.607 | 0.282 | 0.435 | 26.81 |
| 25600 | -2 | -0.00002 | 0.549 | 0.610 | 0.607 | 0.279 | 0.433 | 26.79 |
| 25600 | -3 | -0.00002 | 0.548 | 0.601 | 0.607 | 0.277 | 0.432 | 26.76 |
| 25600 | 0 | -0.00002 | 0.549 | 0.642 | 0.606 | 0.289 | 0.437 | 26.83 |
| 3200 | -1 | -0.00002 | 0.549 | 0.624 | 0.726 | 0.363 | 0.470 | 26.87 |
| 3200 | -2 | -0.00002 | 0.549 | 0.610 | 0.726 | 0.357 | 0.467 | 26.84 |
| 3200 | -3 | -0.00002 | 0.549 | 0.600 | 0.726 | 0.353 | 0.465 | 26.77 |
| 3200 | 0 | -0.00002 | 0.548 | 0.641 | 0.726 | 0.375 | 0.474 | 26.9 |
| 400 | -1 | -0.00002 | 0.548 | 0.623 | 0.787 | 0.412 | 0.490 | 26.94 |
| 400 | -3 | -0.00002 | 0.549 | 0.600 | 0.787 | 0.394 | 0.483 | 26.82 |
| 400 | 0 | -0.00002 | 0.548 | 0.641 | 0.787 | 0.429 | 0.497 | 27.04 |
| 6400 | -1 | -0.00002 | 0.549 | 0.624 | 0.697 | 0.343 | 0.461 | 26.87 |
| 6400 | -2 | -0.00002 | 0.549 | 0.610 | 0.697 | 0.337 | 0.458 | 26.81 |
| 6400 | -3 | -0.00002 | 0.549 | 0.601 | 0.697 | 0.334 | 0.457 | 26.81 |
| 6400 | 0 | -0.00002 | 0.549 | 0.641 | 0.697 | 0.352 | 0.465 | 26.89 |
| 800 | -1 | -0.00002 | 0.548 | 0.623 | 0.770 | 0.397 | 0.484 | 26.88 |
| 800 | -2 | -0.00002 | 0.548 | 0.610 | 0.770 | 0.388 | 0.480 | 26.85 |
| 800 | -3 | -0.00002 | 0.548 | 0.600 | 0.770 | 0.382 | 0.477 | 26.86 |
| 800 | 0 | -0.00002 | 0.548 | 0.641 | 0.770 | 0.413 | 0.490 | 27 |

Table 1: BLOOM-560M results

| $T_{\text{sum}}$ | $T_{\text{AS}}$ | $T_{\text{QP}}$ | $P_{\text{QP}}$ | $P_{\text{AS}}$ | $P_{\text{sum}}$ | $P_{\text{sm}}$ | $P_{\text{NAQ}}$ | Perplexity |
|---|---|---|---|---|---|---|---|---|
| 12800 | -1 | -0.00002 | 0.515 | 0.600 | 0.713 | 0.344 | 0.442 | 22.56 |
| 12800 | -2 | -0.00002 | 0.515 | 0.584 | 0.713 | 0.336 | 0.438 | 22.47 |
| 12800 | -3 | -0.00002 | 0.515 | 0.573 | 0.713 | 0.332 | 0.436 | 22.46 |
| 12800 | 0 | -0.00002 | 0.515 | 0.623 | 0.713 | 0.357 | 0.447 | 22.6 |
| 1600 | -1 | -0.00002 | 0.514 | 0.600 | 0.814 | 0.420 | 0.474 | 22.63 |
| 1600 | -2 | -0.00002 | 0.514 | 0.585 | 0.813 | 0.409 | 0.469 | 22.57 |
| 1600 | -2 | -0.00002 | 0.514 | 0.585 | 0.813 | 0.409 | 0.469 | 22.58 |
| 1600 | -3 | -0.00002 | 0.514 | 0.573 | 0.813 | 0.402 | 0.466 | 22.54 |
| 1600 | 0 | -0.00002 | 0.514 | 0.622 | 0.813 | 0.438 | 0.482 | 22.82 |
| 200 | -1 | -0.00002 | 0.515 | 0.599 | 0.864 | 0.464 | 0.493 | 22.79 |
| 200 | -3 | -0.00002 | 0.514 | 0.573 | 0.864 | 0.440 | 0.483 | 22.6 |
| 200 | 0 | -0.00002 | 0.515 | 0.620 | 0.865 | 0.486 | 0.502 | 23.09 |
| 25600 | -1 | -0.00002 | 0.515 | 0.600 | 0.641 | 0.296 | 0.421 | 22.53 |
| 25600 | -2 | -0.00002 | 0.515 | 0.585 | 0.642 | 0.290 | 0.419 | 22.49 |
| 25600 | -3 | -0.00002 | 0.514 | 0.573 | 0.642 | 0.286 | 0.417 | 22.44 |
| 25600 | 0 | -0.00002 | 0.515 | 0.622 | 0.642 | 0.307 | 0.426 | 22.59 |
| 3200 | -1 | -0.00002 | 0.515 | 0.600 | 0.791 | 0.402 | 0.466 | 22.61 |
| 3200 | -2 | -0.00002 | 0.514 | 0.585 | 0.791 | 0.392 | 0.462 | 22.51 |
| 3200 | -3 | -0.00002 | 0.515 | 0.573 | 0.791 | 0.386 | 0.460 | 22.47 |
| 3200 | 0 | -0.00002 | 0.515 | 0.622 | 0.790 | 0.418 | 0.474 | 22.7 |
| 400 | -1 | -0.00002 | 0.514 | 0.599 | 0.850 | 0.450 | 0.487 | 22.75 |
| 400 | -3 | -0.00002 | 0.514 | 0.573 | 0.849 | 0.428 | 0.478 | 22.54 |
| 400 | 0 | -0.00002 | 0.514 | 0.620 | 0.850 | 0.471 | 0.496 | 23.02 |
| 6400 | -1 | -0.00002 | 0.514 | 0.600 | 0.760 | 0.378 | 0.456 | 22.55 |
| 6400 | -2 | -0.00002 | 0.514 | 0.584 | 0.760 | 0.369 | 0.452 | 22.45 |
| 6400 | -3 | -0.00002 | 0.514 | 0.573 | 0.760 | 0.364 | 0.450 | 22.53 |
| 6400 | 0 | -0.00002 | 0.514 | 0.622 | 0.760 | 0.393 | 0.462 | 22.57 |
| 800 | -1 | -0.00002 | 0.514 | 0.599 | 0.833 | 0.435 | 0.481 | 22.71 |
| 800 | -2 | -0.00002 | 0.515 | 0.584 | 0.833 | 0.423 | 0.476 | 22.66 |
| 800 | -3 | -0.00002 | 0.515 | 0.573 | 0.832 | 0.416 | 0.472 | 22.52 |
| 800 | 0 | -0.00002 | 0.515 | 0.621 | 0.833 | 0.455 | 0.489 | 22.99 |

Table 2: BLOOM-1B1 results

| $T_{\text{sum}}$ | $T_{\text{AS}}$ | $T_{\text{QP}}$ | $P_{\text{QP}}$ | $P_{\text{AS}}$ | $P_{\text{sum}}$ | $P_{\text{sm}}$ | $P_{\text{NAQ}}$ | Perplexity |
|---|---|---|---|---|---|---|---|---|
| 12800 | -3 | 0 | 0.917 | 0.819 | 0.091 | 0.054 | 0.411 | 30.27 |
| 1600 | -3 | 0 | 0.917 | 0.820 | 0.221 | 0.137 | 0.460 | 30.12 |
| 200 | -3 | 0 | 0.917 | 0.817 | 0.479 | 0.329 | 0.572 | 30.48 |
| 25600 | -3 | 0 | 0.917 | 0.818 | 0.067 | 0.039 | 0.402 | 30.59 |
| 3200 | -3 | 0 | 0.916 | 0.817 | 0.165 | 0.101 | 0.439 | 30.23 |
| 400 | -3 | 0 | 0.917 | 0.815 | 0.386 | 0.253 | 0.527 | 30.16 |
| 6400 | -3 | 0 | 0.916 | 0.817 | 0.122 | 0.074 | 0.422 | 29.82 |
| 800 | -3 | 0 | 0.916 | 0.818 | 0.300 | 0.190 | 0.491 | 30.5 |
| 12800 | -2 | 0 | 0.917 | 0.866 | 0.090 | 0.059 | 0.414 | 29.53 |
| 1600 | -2 | 0 | 0.917 | 0.865 | 0.223 | 0.151 | 0.468 | 29.88 |
| 200 | -2 | 0 | 0.916 | 0.865 | 0.474 | 0.355 | 0.587 | 30.04 |
| 25600 | -2 | 0 | 0.917 | 0.865 | 0.067 | 0.042 | 0.404 | 30.16 |
| 3200 | -2 | 0 | 0.917 | 0.866 | 0.161 | 0.109 | 0.443 | 30.11 |
| 400 | -2 | 0 | 0.917 | 0.866 | 0.378 | 0.272 | 0.539 | 30.22 |
| 6400 | -2 | 0 | 0.916 | 0.865 | 0.122 | 0.081 | 0.427 | 30.35 |
| 800 | -2 | 0 | 0.917 | 0.865 | 0.297 | 0.206 | 0.500 | 30.07 |
| 12800 | -1 | 0 | 0.916 | 0.908 | 0.091 | 0.064 | 0.417 | 29.75 |
| 1600 | -1 | 0 | 0.916 | 0.908 | 0.225 | 0.168 | 0.477 | 30.25 |
| 200 | -1 | 0 | 0.917 | 0.909 | 0.476 | 0.391 | 0.609 | 31.51 |
| 25600 | -1 | 0 | 0.917 | 0.909 | 0.067 | 0.046 | 0.406 | 30.33 |
| 3200 | -1 | 0 | 0.916 | 0.909 | 0.162 | 0.120 | 0.449 | 30.07 |
| 400 | -1 | 0 | 0.916 | 0.909 | 0.379 | 0.300 | 0.555 | 30.21 |
| 6400 | -1 | 0 | 0.917 | 0.908 | 0.122 | 0.088 | 0.431 | 30.07 |
| 800 | -1 | 0 | 0.916 | 0.909 | 0.296 | 0.226 | 0.512 | 30.64 |
| 12800 | 0 | 0 | 0.917 | 0.946 | 0.091 | 0.070 | 0.420 | 30.07 |
| 1600 | 0 | 0 | 0.916 | 0.946 | 0.223 | 0.182 | 0.486 | 30.26 |
| 200 | 0 | 0 | 0.917 | 0.945 | 0.474 | 0.421 | 0.626 | 33.09 |
| 25600 | 0 | 0 | 0.917 | 0.946 | 0.068 | 0.051 | 0.409 | 29.83 |
| 3200 | 0 | 0 | 0.917 | 0.945 | 0.165 | 0.132 | 0.457 | 30.03 |
| 400 | 0 | 0 | 0.917 | 0.945 | 0.384 | 0.332 | 0.574 | 32.26 |
| 6400 | 0 | 0 | 0.916 | 0.945 | 0.120 | 0.094 | 0.434 | 30.55 |
| 800 | 0 | 0 | 0.916 | 0.944 | 0.297 | 0.249 | 0.525 | 31.53 |

Table 3: OPT-125M results

| $T_{\text{sum}}$ | $T_{\text{AS}}$ | $T_{\text{QP}}$ | $P_{\text{QP}}$ | $P_{\text{AS}}$ | $P_{\text{sum}}$ | $P_{\text{sm}}$ | $P_{\text{NAQ}}$ | Perplexity |
|---|---|---|---|---|---|---|---|---|
| 200 | -3 | 0 | 0.891 | 0.881 | 0.355 | 0.272 | 0.547 | 22.13 |
| 400 | -3 | 0 | 0.891 | 0.881 | 0.260 | 0.192 | 0.503 | 22.19 |
| 800 | -3 | 0 | 0.891 | 0.881 | 0.185 | 0.132 | 0.469 | 22.17 |
| 1600 | -3 | 0 | 0.891 | 0.880 | 0.124 | 0.089 | 0.446 | 22.14 |
| 3200 | -3 | 0 | 0.891 | 0.881 | 0.081 | 0.063 | 0.431 | 22.14 |
| 6400 | -3 | 0 | 0.891 | 0.881 | 0.060 | 0.047 | 0.422 | 22.15 |
| 12800 | -3 | 0 | 0.892 | 0.881 | 0.046 | 0.036 | 0.416 | 22.18 |
| 25600 | -3 | 0 | 0.891 | 0.881 | 0.037 | 0.029 | 0.412 | 22.13 |
| 200 | -2 | 0 | 0.891 | 0.918 | 0.353 | 0.288 | 0.556 | 22.19 |
| 400 | -2 | 0 | 0.891 | 0.917 | 0.259 | 0.203 | 0.509 | 22.16 |
| 800 | -2 | 0 | 0.891 | 0.917 | 0.186 | 0.140 | 0.474 | 22.22 |
| 1600 | -2 | 0 | 0.891 | 0.917 | 0.124 | 0.093 | 0.448 | 22.18 |
| 3200 | -2 | 0 | 0.891 | 0.916 | 0.080 | 0.065 | 0.432 | 22.22 |
| 6400 | -2 | 0 | 0.891 | 0.917 | 0.060 | 0.048 | 0.423 | 22.13 |
| 12800 | -2 | 0 | 0.891 | 0.916 | 0.046 | 0.037 | 0.417 | 22.18 |
| 25600 | -2 | 0 | 0.891 | 0.916 | 0.037 | 0.030 | 0.412 | 22.15 |
| 200 | -1 | 0 | 0.891 | 0.943 | 0.354 | 0.304 | 0.565 | 22.2 |
| 400 | -1 | 0 | 0.891 | 0.942 | 0.259 | 0.213 | 0.514 | 22.18 |
| 800 | -1 | 0 | 0.892 | 0.942 | 0.185 | 0.146 | 0.477 | 22.16 |
| 1600 | -1 | 0 | 0.891 | 0.942 | 0.122 | 0.097 | 0.450 | 22.15 |
| 3200 | -1 | 0 | 0.891 | 0.942 | 0.080 | 0.068 | 0.434 | 22.13 |
| 6400 | -1 | 0 | 0.891 | 0.942 | 0.059 | 0.050 | 0.424 | 22.19 |
| 12800 | -1 | 0 | 0.891 | 0.942 | 0.046 | 0.038 | 0.417 | 22.16 |
| 25600 | -1 | 0 | 0.891 | 0.942 | 0.037 | 0.030 | 0.413 | 22.14 |
| 200 | 0 | 0 | 0.891 | 0.965 | 0.352 | 0.320 | 0.574 | 22.34 |
| 400 | 0 | 0 | 0.892 | 0.965 | 0.260 | 0.229 | 0.524 | 22.26 |
| 800 | 0 | 0 | 0.891 | 0.964 | 0.185 | 0.158 | 0.484 | 22.22 |
| 1600 | 0 | 0 | 0.892 | 0.964 | 0.124 | 0.104 | 0.454 | 22.14 |
| 3200 | 0 | 0 | 0.891 | 0.964 | 0.080 | 0.070 | 0.435 | 22.18 |
| 6400 | 0 | 0 | 0.892 | 0.964 | 0.059 | 0.051 | 0.425 | 22.13 |
| 12800 | 0 | 0 | 0.891 | 0.964 | 0.046 | 0.039 | 0.418 | 22.13 |
| 25600 | 0 | 0 | 0.891 | 0.964 | 0.037 | 0.031 | 0.413 | 22.2 |

Table 4: OPT-350M results

| $T_{\text{sum}}$ | $T_{\text{AS}}$ | $T_{\text{QP}}$ | $P_{\text{QP}}$ | $P_{\text{AS}}$ | $P_{\text{sum}}$ | $P_{\text{sm}}$ | $P_{\text{NAQ}}$ | Accuracy |
|---|---|---|---|---|---|---|---|---|
| 200 | 0 | -0.0002 | 0.413 | 0.666 | 0.626 | 0.333 | 0.373 | 70.3 |
| 12800 | -3 | -0.0002 | 0.413 | 0.126 | 0.099 | 0.034 | 0.222 | 70.1 |
| 12800 | -2 | -0.0002 | 0.413 | 0.256 | 0.099 | 0.041 | 0.226 | 70.0 |
| 25600 | 0 | -0.0002 | 0.413 | 0.670 | 0.083 | 0.048 | 0.229 | 70.0 |
| 3200 | -1 | -0.0002 | 0.413 | 0.458 | 0.142 | 0.061 | 0.236 | 70.0 |
| 6400 | -1 | -0.0002 | 0.413 | 0.456 | 0.117 | 0.054 | 0.232 | 70.0 |
| 1600 | -3 | -0.0002 | 0.413 | 0.127 | 0.189 | 0.044 | 0.227 | 69.9 |
| 400 | -2 | -0.0002 | 0.414 | 0.262 | 0.450 | 0.094 | 0.253 | 69.9 |
| 6400 | -3 | -0.0002 | 0.413 | 0.126 | 0.115 | 0.037 | 0.224 | 69.9 |
| 6400 | 0 | -0.0002 | 0.412 | 0.672 | 0.115 | 0.063 | 0.237 | 69.9 |
| 400 | -1 | -0.0002 | 0.413 | 0.464 | 0.446 | 0.149 | 0.280 | 69.8 |
| 800 | 0 | -0.0002 | 0.412 | 0.676 | 0.284 | 0.135 | 0.272 | 69.8 |
| 12800 | -1 | -0.0002 | 0.413 | 0.461 | 0.097 | 0.048 | 0.229 | 69.7 |
| 12800 | 0 | -0.0002 | 0.413 | 0.674 | 0.097 | 0.055 | 0.233 | 69.7 |
| 200 | -2 | -0.0002 | 0.412 | 0.260 | 0.607 | 0.122 | 0.266 | 69.7 |
| 200 | -3 | -0.0002 | 0.413 | 0.125 | 0.618 | 0.069 | 0.240 | 69.7 |
| 3200 | -2 | -0.0002 | 0.413 | 0.259 | 0.139 | 0.050 | 0.230 | 69.7 |
| 400 | 0 | -0.0002 | 0.413 | 0.673 | 0.451 | 0.223 | 0.317 | 69.7 |
| 800 | -1 | -0.0002 | 0.413 | 0.465 | 0.287 | 0.101 | 0.256 | 69.7 |
| 800 | -2 | -0.0002 | 0.413 | 0.259 | 0.289 | 0.071 | 0.241 | 69.7 |
| 800 | -3 | -0.0002 | 0.413 | 0.125 | 0.287 | 0.050 | 0.230 | 69.7 |
| 1600 | -2 | -0.0002 | 0.414 | 0.260 | 0.184 | 0.056 | 0.234 | 69.6 |
| 25600 | -1 | -0.0002 | 0.413 | 0.461 | 0.081 | 0.042 | 0.226 | 69.6 |
| 25600 | -3 | -0.0002 | 0.413 | 0.125 | 0.083 | 0.032 | 0.221 | 69.6 |
| 6400 | -2 | -0.0002 | 0.413 | 0.259 | 0.114 | 0.045 | 0.228 | 69.6 |
| 1600 | -1 | -0.0002 | 0.415 | 0.454 | 0.197 | 0.074 | 0.243 | 69.5 |
| 1600 | 0 | -0.0002 | 0.412 | 0.677 | 0.186 | 0.092 | 0.251 | 69.5 |
| 25600 | -2 | -0.0002 | 0.413 | 0.259 | 0.082 | 0.036 | 0.224 | 69.5 |
| 3200 | -3 | -0.0002 | 0.413 | 0.126 | 0.136 | 0.040 | 0.225 | 69.5 |
| 3200 | 0 | -0.0002 | 0.411 | 0.680 | 0.138 | 0.073 | 0.241 | 69.4 |
| 400 | -3 | -0.0002 | 0.413 | 0.128 | 0.441 | 0.059 | 0.234 | 69.4 |
| 200 | -1 | -0.0002 | 0.411 | 0.470 | 0.597 | 0.208 | 0.309 | 69.2 |

Table 5: deit_tiny_patch16_224 results

| $T_{\text{sum}}$ | $T_{\text{AS}}$ | $T_{\text{QP}}$ | $P_{\text{QP}}$ | $P_{\text{AS}}$ | $P_{\text{sum}}$ | $P_{\text{sm}}$ | $P_{\text{NAQ}}$ | Accuracy |
|---|---|---|---|---|---|---|---|---|
| 400 | 0 | -0.0002 | 0.428 | 0.685 | 0.425 | 0.219 | 0.330 | 79.4 |
| 6400 | -3 | -0.0002 | 0.426 | 0.170 | 0.098 | 0.023 | 0.237 | 79.3 |
| 12800 | -1 | -0.0002 | 0.426 | 0.489 | 0.074 | 0.030 | 0.240 | 79.2 |
| 12800 | -3 | -0.0002 | 0.426 | 0.172 | 0.074 | 0.020 | 0.236 | 79.2 |
| 1600 | -1 | -0.0002 | 0.426 | 0.490 | 0.176 | 0.067 | 0.258 | 79.2 |
| 1600 | -2 | -0.0002 | 0.426 | 0.304 | 0.179 | 0.049 | 0.249 | 79.2 |
| 200 | -1 | -0.0002 | 0.426 | 0.488 | 0.614 | 0.222 | 0.330 | 79.2 |
| 200 | -3 | -0.0002 | 0.426 | 0.172 | 0.614 | 0.068 | 0.258 | 79.2 |
| 200 | 0 | -0.0002 | 0.429 | 0.683 | 0.626 | 0.348 | 0.391 | 79.2 |
| 3200 | -1 | -0.0002 | 0.428 | 0.483 | 0.130 | 0.050 | 0.251 | 79.2 |
| 3200 | -3 | -0.0002 | 0.425 | 0.174 | 0.128 | 0.027 | 0.239 | 79.2 |
| 400 | -1 | -0.0002 | 0.425 | 0.491 | 0.413 | 0.150 | 0.296 | 79.2 |
| 400 | -2 | -0.0002 | 0.426 | 0.301 | 0.415 | 0.094 | 0.271 | 79.2 |
| 6400 | -2 | -0.0002 | 0.425 | 0.304 | 0.096 | 0.030 | 0.240 | 79.2 |
| 800 | -1 | -0.0002 | 0.426 | 0.487 | 0.266 | 0.099 | 0.272 | 79.2 |
| 800 | -3 | -0.0002 | 0.426 | 0.171 | 0.270 | 0.043 | 0.246 | 79.2 |
| 12800 | -2 | -0.0002 | 0.425 | 0.303 | 0.074 | 0.024 | 0.237 | 79.1 |
| 12800 | 0 | -0.0002 | 0.426 | 0.689 | 0.074 | 0.036 | 0.244 | 79.1 |
| 1600 | -3 | -0.0002 | 0.428 | 0.172 | 0.174 | 0.033 | 0.243 | 79.1 |
| 200 | -2 | -0.0002 | 0.427 | 0.299 | 0.616 | 0.125 | 0.286 | 79.1 |
| 25600 | -3 | -0.0002 | 0.424 | 0.175 | 0.058 | 0.017 | 0.233 | 79.1 |
| 25600 | 0 | -0.0002 | 0.426 | 0.692 | 0.059 | 0.029 | 0.240 | 79.1 |
| 400 | -3 | -0.0002 | 0.426 | 0.171 | 0.411 | 0.055 | 0.252 | 79.1 |
| 6400 | -1 | -0.0002 | 0.427 | 0.489 | 0.096 | 0.038 | 0.245 | 79.1 |
| 800 | -2 | -0.0002 | 0.424 | 0.303 | 0.266 | 0.068 | 0.257 | 79.1 |
| 1600 | 0 | -0.0002 | 0.426 | 0.688 | 0.178 | 0.088 | 0.268 | 79.0 |
| 25600 | -1 | -0.0002 | 0.425 | 0.491 | 0.059 | 0.024 | 0.237 | 79.0 |
| 3200 | 0 | -0.0002 | 0.426 | 0.692 | 0.127 | 0.062 | 0.256 | 79.0 |
| 6400 | 0 | -0.0002 | 0.427 | 0.693 | 0.093 | 0.046 | 0.248 | 79.0 |
| 25600 | -2 | -0.0002 | 0.426 | 0.304 | 0.058 | 0.020 | 0.236 | 78.9 |
| 3200 | -2 | -0.0002 | 0.425 | 0.300 | 0.121 | 0.034 | 0.242 | 78.9 |
| 800 | 0 | -0.0002 | 0.427 | 0.694 | 0.260 | 0.130 | 0.287 | 78.8 |

Table 6: deit_small_patch16_224 results

| $T_{\text{sum}}$ | $T_{\text{AS}}$ | $T_{\text{QP}}$ | $P_{\text{QP}}$ | $P_{\text{AS}}$ | $P_{\text{sum}}$ | $P_{\text{sm}}$ | $P_{\text{NAQ}}$ | Accuracy |
|---|---|---|---|---|---|---|---|---|
| 12800 | -1 | -0.00002 | 0.531 | 0.502 | 0.124 | 0.043 | 0.285 | 72.6 |
| 12800 | -2 | -0.00002 | 0.530 | 0.336 | 0.124 | 0.034 | 0.280 | 72.9 |
| 12800 | -3 | -0.00002 | 0.530 | 0.209 | 0.123 | 0.026 | 0.276 | 71.0 |
| 12800 | 0 | -0.00002 | 0.530 | 0.666 | 0.122 | 0.054 | 0.290 | 72.9 |
| 1600 | -1 | -0.00002 | 0.530 | 0.502 | 0.266 | 0.087 | 0.307 | 73.5 |
| 1600 | -2 | -0.00002 | 0.530 | 0.336 | 0.264 | 0.059 | 0.293 | 72.6 |
| 1600 | -3 | -0.00002 | 0.531 | 0.208 | 0.262 | 0.039 | 0.283 | 72.8 |
| 1600 | 0 | -0.00002 | 0.529 | 0.662 | 0.265 | 0.118 | 0.322 | 72.2 |
| 200 | -1 | -0.00002 | 0.530 | 0.497 | 0.693 | 0.261 | 0.394 | 72.8 |
| 200 | -2 | -0.00002 | 0.529 | 0.334 | 0.690 | 0.159 | 0.343 | 72.4 |
| 200 | -3 | -0.00002 | 0.530 | 0.207 | 0.687 | 0.090 | 0.308 | 73.4 |
| 200 | 0 | -0.00002 | 0.530 | 0.662 | 0.691 | 0.379 | 0.454 | 72.9 |
| 25600 | -1 | -0.00002 | 0.530 | 0.502 | 0.106 | 0.038 | 0.282 | 69.7 |
| 25600 | -2 | -0.00002 | 0.530 | 0.335 | 0.104 | 0.030 | 0.278 | 72.6 |
| 25600 | -3 | -0.00002 | 0.530 | 0.202 | 0.106 | 0.024 | 0.275 | 72.6 |
| 25600 | 0 | -0.00002 | 0.531 | 0.667 | 0.104 | 0.046 | 0.287 | 73.0 |
| 3200 | -1 | -0.00002 | 0.529 | 0.498 | 0.192 | 0.064 | 0.295 | 72.6 |
| 3200 | -2 | -0.00002 | 0.530 | 0.335 | 0.192 | 0.046 | 0.286 | 72.9 |
| 3200 | -3 | -0.00002 | 0.530 | 0.202 | 0.194 | 0.033 | 0.279 | 72.8 |
| 3200 | 0 | -0.00002 | 0.530 | 0.665 | 0.192 | 0.084 | 0.305 | 72.4 |
| 400 | -1 | -0.00002 | 0.529 | 0.501 | 0.519 | 0.180 | 0.354 | 72.2 |
| 400 | -2 | -0.00002 | 0.531 | 0.339 | 0.522 | 0.114 | 0.321 | 73.1 |
| 400 | -3 | -0.00002 | 0.531 | 0.211 | 0.521 | 0.065 | 0.296 | 71.8 |
| 400 | 0 | -0.00002 | 0.530 | 0.667 | 0.527 | 0.267 | 0.397 | 72.5 |
| 6400 | -1 | -0.00002 | 0.530 | 0.500 | 0.149 | 0.051 | 0.289 | 73.5 |
| 6400 | -2 | -0.00002 | 0.529 | 0.333 | 0.149 | 0.039 | 0.282 | 72.6 |
| 6400 | -3 | -0.00002 | 0.530 | 0.208 | 0.149 | 0.029 | 0.278 | 73.1 |
| 6400 | 0 | -0.00002 | 0.529 | 0.664 | 0.149 | 0.065 | 0.295 | 72.6 |
| 800 | -1 | -0.00002 | 0.530 | 0.499 | 0.371 | 0.122 | 0.325 | 72.4 |
| 800 | -2 | -0.00002 | 0.530 | 0.333 | 0.376 | 0.080 | 0.303 | 72.8 |
| 800 | -3 | -0.00002 | 0.530 | 0.205 | 0.374 | 0.049 | 0.288 | 72.6 |
| 800 | 0 | -0.00002 | 0.531 | 0.669 | 0.367 | 0.172 | 0.351 | 72.4 |

Table 7: vit_tiny_patch16_224 results

| $T_{\text{sum}}$ | $T_{\text{AS}}$ | $T_{\text{QP}}$ | $P_{\text{QP}}$ | $P_{\text{AS}}$ | $P_{\text{sum}}$ | $P_{\text{sm}}$ | $P_{\text{NAQ}}$ | Accuracy |
|---|---|---|---|---|---|---|---|---|
| 12800 | -1 | -0.00002 | 0.561 | 0.531 | 0.078 | 0.031 | 0.313 | 80.5 |
| 12800 | -2 | -0.00002 | 0.561 | 0.363 | 0.078 | 0.026 | 0.310 | 80.6 |
| 12800 | -3 | -0.00002 | 0.562 | 0.232 | 0.078 | 0.021 | 0.308 | 80.4 |
| 12800 | 0 | -0.00002 | 0.561 | 0.700 | 0.078 | 0.037 | 0.316 | 80.7 |
| 1600 | -1 | -0.00002 | 0.561 | 0.530 | 0.198 | 0.070 | 0.331 | 80.5 |
| 1600 | -2 | -0.00002 | 0.561 | 0.364 | 0.200 | 0.051 | 0.322 | 80.5 |
| 1600 | -3 | -0.00002 | 0.561 | 0.230 | 0.199 | 0.035 | 0.315 | 80.5 |
| 1600 | 0 | -0.00002 | 0.561 | 0.700 | 0.198 | 0.093 | 0.341 | 80.5 |
| 200 | -1 | -0.00002 | 0.561 | 0.518 | 0.636 | 0.247 | 0.414 | 80.5 |
| 200 | -2 | -0.00002 | 0.561 | 0.364 | 0.608 | 0.151 | 0.369 | 80.6 |
| 200 | -3 | -0.00002 | 0.561 | 0.229 | 0.612 | 0.091 | 0.341 | 80.7 |
| 200 | 0 | -0.00002 | 0.562 | 0.693 | 0.626 | 0.354 | 0.464 | 80.6 |
| 25600 | -1 | -0.00002 | 0.562 | 0.531 | 0.065 | 0.027 | 0.311 | 80.7 |
| 25600 | -2 | -0.00002 | 0.561 | 0.359 | 0.065 | 0.023 | 0.309 | 80.4 |
| 25600 | -3 | -0.00002 | 0.562 | 0.228 | 0.064 | 0.019 | 0.307 | 80.6 |
| 25600 | 0 | -0.00002 | 0.561 | 0.708 | 0.064 | 0.031 | 0.313 | 80.6 |
| 3200 | -1 | -0.00002 | 0.561 | 0.532 | 0.135 | 0.049 | 0.321 | 80.6 |
| 3200 | -2 | -0.00002 | 0.561 | 0.364 | 0.136 | 0.038 | 0.316 | 80.6 |
| 3200 | -3 | -0.00002 | 0.561 | 0.234 | 0.132 | 0.028 | 0.311 | 80.5 |
| 3200 | 0 | -0.00002 | 0.561 | 0.697 | 0.136 | 0.063 | 0.327 | 80.6 |
| 400 | -1 | -0.00002 | 0.561 | 0.532 | 0.441 | 0.161 | 0.374 | 80.4 |
| 400 | -2 | -0.00002 | 0.561 | 0.363 | 0.447 | 0.106 | 0.348 | 80.6 |
| 400 | -3 | -0.00002 | 0.561 | 0.230 | 0.445 | 0.065 | 0.329 | 80.5 |
| 400 | 0 | -0.00002 | 0.561 | 0.692 | 0.459 | 0.238 | 0.410 | 80.4 |
| 6400 | -1 | -0.00002 | 0.561 | 0.532 | 0.099 | 0.038 | 0.316 | 80.7 |
| 6400 | -2 | -0.00002 | 0.561 | 0.362 | 0.099 | 0.031 | 0.312 | 80.4 |
| 6400 | -3 | -0.00002 | 0.562 | 0.231 | 0.098 | 0.024 | 0.310 | 80.6 |
| 6400 | 0 | -0.00002 | 0.561 | 0.708 | 0.098 | 0.046 | 0.320 | 80.4 |
| 800 | -1 | -0.00002 | 0.561 | 0.530 | 0.305 | 0.107 | 0.348 | 80.6 |
| 800 | -2 | -0.00002 | 0.561 | 0.365 | 0.302 | 0.072 | 0.332 | 80.6 |
| 800 | -3 | -0.00002 | 0.561 | 0.230 | 0.305 | 0.047 | 0.320 | 80.6 |
| 800 | 0 | -0.00002 | 0.562 | 0.695 | 0.311 | 0.151 | 0.369 | 80.5 |

Table 8: vit_small_patch16_224 results

| $T_{\text{sum}}$ | $T_{\text{AS}}$ | $T_{\text{QP}}$ | $P_{\text{QP}}$ | $P_{\text{AS}}$ | $P_{\text{sum}}$ | $P_{\text{sm}}$ | $P_{\text{NAQ}}$ | Accuracy |
|---|---|---|---|---|---|---|---|---|
| 12800 | -1 | -0.00002 | 0.530 | 0.477 | 0.024 | 0.008 | 0.300 | 74.9 |
| 12800 | -2 | -0.00002 | 0.530 | 0.291 | 0.024 | 0.006 | 0.300 | 75.1 |
| 12800 | -3 | -0.00002 | 0.529 | 0.158 | 0.024 | 0.005 | 0.298 | 74.9 |
| 12800 | 0 | -0.00002 | 0.529 | 0.671 | 0.024 | 0.011 | 0.301 | 75.0 |
| 1600 | -1 | -0.00002 | 0.530 | 0.475 | 0.075 | 0.024 | 0.307 | 74.9 |
| 1600 | -2 | -0.00002 | 0.530 | 0.287 | 0.076 | 0.017 | 0.304 | 74.8 |
| 1600 | -3 | -0.00002 | 0.530 | 0.159 | 0.075 | 0.012 | 0.302 | 75.0 |
| 1600 | 0 | -0.00002 | 0.530 | 0.671 | 0.075 | 0.032 | 0.311 | 75.1 |
| 200 | -1 | -0.00002 | 0.529 | 0.467 | 0.278 | 0.091 | 0.337 | 75.1 |
| 200 | -2 | -0.00002 | 0.530 | 0.293 | 0.266 | 0.056 | 0.321 | 74.9 |
| 200 | -3 | -0.00002 | 0.530 | 0.156 | 0.276 | 0.034 | 0.312 | 75.0 |
| 200 | 0 | -0.00002 | 0.530 | 0.672 | 0.272 | 0.129 | 0.353 | 75.0 |
| 25600 | -1 | -0.00002 | 0.530 | 0.472 | 0.017 | 0.006 | 0.299 | 75.0 |
| 25600 | -2 | -0.00002 | 0.530 | 0.290 | 0.017 | 0.004 | 0.299 | 74.9 |
| 25600 | -3 | -0.00002 | 0.530 | 0.159 | 0.017 | 0.003 | 0.298 | 75.0 |
| 25600 | 0 | -0.00002 | 0.529 | 0.670 | 0.017 | 0.007 | 0.300 | 74.9 |
| 3200 | -1 | -0.00002 | 0.530 | 0.473 | 0.050 | 0.017 | 0.304 | 74.9 |
| 3200 | -2 | -0.00002 | 0.529 | 0.289 | 0.051 | 0.013 | 0.302 | 74.9 |
| 3200 | -3 | -0.00002 | 0.530 | 0.157 | 0.051 | 0.009 | 0.301 | 75.0 |
| 3200 | 0 | -0.00002 | 0.529 | 0.666 | 0.051 | 0.022 | 0.306 | 75.1 |
| 400 | -1 | -0.00002 | 0.530 | 0.475 | 0.169 | 0.055 | 0.321 | 75.0 |
| 400 | -2 | -0.00002 | 0.529 | 0.288 | 0.172 | 0.037 | 0.313 | 74.9 |
| 400 | -3 | -0.00002 | 0.529 | 0.156 | 0.171 | 0.023 | 0.306 | 75.1 |
| 400 | 0 | -0.00002 | 0.530 | 0.670 | 0.172 | 0.078 | 0.331 | 74.7 |
| 6400 | -1 | -0.00002 | 0.529 | 0.470 | 0.035 | 0.012 | 0.302 | 74.8 |
| 6400 | -2 | -0.00002 | 0.530 | 0.291 | 0.035 | 0.009 | 0.301 | 75.1 |
| 6400 | -3 | -0.00002 | 0.530 | 0.159 | 0.035 | 0.006 | 0.299 | 74.9 |
| 6400 | 0 | -0.00002 | 0.530 | 0.671 | 0.034 | 0.015 | 0.303 | 74.9 |
| 800 | -1 | -0.00002 | 0.529 | 0.468 | 0.115 | 0.036 | 0.313 | 74.9 |
| 800 | -2 | -0.00002 | 0.530 | 0.290 | 0.112 | 0.024 | 0.307 | 75.0 |
| 800 | -3 | -0.00002 | 0.530 | 0.159 | 0.111 | 0.016 | 0.304 | 75.0 |
| 800 | 0 | -0.00002 | 0.530 | 0.673 | 0.112 | 0.048 | 0.318 | 74.9 |

Table 9: vit_small_patch32_224 results