# OpenReview forum: "NAQ: Nonlinearity-Aware Quantization"
_ICLR.cc/2025/Conference — Submitted to ICLR 2025_

### Official Review · Reviewer_Uw9k · 2024-10-28

**Soundness:** 3
**Presentation:** 3
**Contribution:** 3
**Rating:** 5
**Confidence:** 4

**Summary:**

This paper focusses on a very important issue related to quantization and proposes a solution with a hypothesis that magnitude of gradient is small for nonlinearities present after activation in MLP layer and attention scores. The Authors propose an interesting method called Nonlinearity-Aware Quantization (NAQ) by computing the FC layer outputs and attention scores at low precision and recomputing the pre-nonlinearity if needed (magnitude of gradient is high) by adding a conditional branch that decides whether re-computation is needed or not.    Results are discussed with an estimation of energy saving calculated using prior work.

**Strengths:**

- The paper is very well written (except few grammatical mistakes as suggested) and focussed on an important research topic.
- Non linear quantization issue is handled in an elegant way where all the computation can be performed in low bit initially and based on the magnitude of the gradient, some of the computation is performed again at higher precision.
- Results suggest that this method has potential to improve quantization of non linear regions.
- This work may have usefulness when a hardware with relevant support is available.

**Weaknesses:**

- One of the main weakness of this paper is lack of results computed on a real hardware (for a obvious reason that such hardware is not available)
- It is perfectly fine to estimate the energy consumption but I think the assumptions are not complete (please see detailed explanations below)

L238 : INT8, FP16 values are used from(Horowitz, 2014) but I am not sure if INT4 energy consumption will follow the same pattern. In my opinion, this is a very large assumption and the results are based on this assumption.

L321: Conditional branch present during inference will pose many challenges on actual hardware. Many edge devices (accelerators) do not support conditional branches and even if it supports, it may add additional complexity. This analysis is ignored during highlighting the benefit of this method.

L372: Also, it is true the Nvidia GPUs support INT4 computation, however, I am still of the opinion that adding conditional branch would add more complexity to the inference which is not studied in this paper due to obvious reasons (no support from the hardware).

L372: It is true that the NVIDIA GPU supports INT4 computation, However, I am still of the opinion that adding conditional branch would add more complexity to the inference which is not studied in this paper due to obvious reasons (no support from the hardware).

L380: (please refer to related question in Questions section ) Why not compare models 1. replacing all FC layers and activation function in the MLP block and 2. vanilla model ? It is important to see the compounding effect of this method on overall models performance.

L492: 10% of energy saving is not very high especially since it is not considering other complexity that this method has added and not considered in energy saving computation.

Grammer :
L121: Use ReLU consistently (RELU and ReLU are interchangeably used)
L368 "there is NOT" -> "There is no"

**Questions:**

Please answer these questions and I will be able to re-visit the score accordingly.

L034 : How about the dynamic range of pre-non linearity values? Does it have high dynamic range and how will it impact the computation of number of bits needed for quantization. Authors can provide an analysis of the dynamic range of pre-nonlinearity values across different models and datasets, and discuss how this impacts their quantization approach.

L238 : INT4 multiply-add cost = 0.065pJ needs a bit more explanation as the numbers from the reference provided does not match. May I request Authors to please provide details of this calculation? Also, INT8, FP16 values are used from(Horowitz, 2014) but I am not sure if INT4 energy consumption will follow the same pattern. can Authors provide additional facts to justify this assumption?

L321: Can Authors please elaborate more on how to handle the conditional branch, any analysis that suggests that the impact will be negligible ?

L372: Can Authors please explain the the relationship of sparsity and NAQ? If the relationship is not established, I would suggest to remove this  reference to improve the clarity of the paper.

L380: My assumption is that comparison is done by adding NAQ only on the first MLP and attention layer, Can Authors please confirm the same? Also, if it is applied to only one layer (first layer), then can Authors provide additional results for other layers and all layers?

L464 : vit_small_patch16_224 energy saving is between 0-10%. Is it because NAQ is applied only on one of the layers ? If this is true, I suggest to expand this study for more layers.

---

### Official Review · Reviewer_FhZz · 2024-10-29

**Soundness:** 2
**Presentation:** 3
**Contribution:** 2
**Rating:** 3
**Confidence:** 4

**Summary:**

This paper proposes a methodology (Nonlinearity Aware Quantization - NAQ) to compute some transformers pre-linearity operations at low precision, with the aim of limiting inference energy consumption. It proposes to leverage regions where the non-linearity gradient is small to perform low precision computations, while recomputing the operations that require high-precision. The targeted operations are the query/key attention product and the MLP FC1. Results are reported on a selection of models (BLOOM, OPT, DeiT, ViT), showing the trade-offs between number of computations avoided (and related energy savings) and perplexity/accuracy.

**Strengths:**

- The topic of improving energy consumption of transformers architectures at inference is of clear interest
- Results show that a significant fraction of computations (in the two targeted operations) can be performed at INT4 with limited performance degradation

**Weaknesses:**

- The main limitation is a lack of hardware demonstration of the proposed concept. Hence, the authors estimate energy consumption from the cost of individual Multiply-Accumulate operations at varying precision (FP16, INT8, INT4). However, using these numbers are not convincing as hardware overheads can vary for different data type and play a significant role
- Estimated advantage against INT8 is limited. As 8-bit formats (FP8, INT8) are becoming more popular along with expanded hardware support, the relevance of the proposed technique diminishes
- As recognized by the authors, this methodology adds inference overhead to latency, so would be only applicable to energy-constrained scenarios
- A key claim of this paper is that some pre-nonlinearity computations can be computed in low precision due to the small gradient of the linearity over some regions of their inputs. For the Query/Key attention computation, the authors use the low-precision quantization product (QP) as a predictor for the gradient, and compare against a manually-selected threshold to determine whether computation in higher precision is needed. However, the correlation between QP and linearity gradient is not established, only briefly mentioned
- The algorithm is not clear. It is stated that the QK Quantized Product is compared element-wise to a threshold. Does *any* QP value above the threshold triggers the high-precision QP recomputation? Would be helpful for the algorithm to be properly spelled out

**Questions:**

1. can the authors comment on their energy consumption estimates and projected energy savings, especially when compared to 8-bit formats?
2. expand on the correlation between QP and MLP FC1 non-linearity gradient
3. clarify the algorithm insofar re-computation is concerned

---

### Official Review · Reviewer_HQ5k · 2024-10-30

**Soundness:** 2
**Presentation:** 2
**Contribution:** 2
**Rating:** 3
**Confidence:** 4

**Summary:**

The paper proposes a novel quantization framework determining whether to quantize FC layers and attention scores dynamically by relying on gradient information from non-linearities.

**Strengths:**

1. The NAQ method leverages the behavior of nonlinear activation functions (specifically, regions of low gradient magnitude) to recompute pre-nonlinearity values, minimizing full precision calculations conditionally. This approach of dynamically deciding whether to quantize or not based on gradient magnitude is interesting.
2. The paper provides a sound theoretical analysis of the proposed framework, highlighting how it differs from existing approaches such as SqueezeLLM.

**Weaknesses:**

1. While NAQ is evaluated on both language and vision models, comparisons with baseline methods are limited, with older large language models (LLMs) and baselines chosen for evaluation.
2. The framework relies on hyperparameters such as quantization thresholds per element. These threshold values determine when to recompute at full precision, but the paper could provide more explicit guidance on setting these values for various use cases or model types.
3. The lack of compatible hardware limits the practical implementation of the proposed method. Additionally, while the authors advocate for improved hardware support for sparsity, this does not directly correlate with the proposed method and could be seen as tangential to the paper’s focus. Lastly, the framework introduces additional latency in model inference due to the dynamic quantization strategy.

**Questions:**

1. Can you share your approach for selecting quantization thresholds, especially for different models?
2. Are there plans to evaluate NAQ on newer models (such as LLama) or against more recent quantization approaches (such as SqueezeLLM)?
3. Can you improve the hardware implementation section with possible directions for implementing such a framework on hardware?
4. Can you evaluate NAQ with different bit-width combinations for both weights and activations? Are there any limitations?

---

### Official Review · Reviewer_GAVm · 2024-11-04

**Soundness:** 2
**Presentation:** 3
**Contribution:** 1
**Rating:** 3
**Confidence:** 5

**Summary:**

This paper proposed a quantization scheme for attention score and FC1/pre-activation computation in transformers, targeting resource-constrained or energy-consumption-sensitive applications. Basic flow is to perform INT4 computations of the two components mentioned above, use the results and the gradient function as estimators for quantization errors, and then perform a partial re-computation in full precision for those elements with higher risks. Based on authors estimate, OPT350m (GPTQ) may have a chance to save ~30% of computation energy while maintaining acceptable model accuracy. ViT (INT8) on ImageNet on the other hand may only have room for ~10% of energy saving.

**Strengths:**

1. Well written and easy to follow.
2. Reasonable amount of experiments to demonstrate and support the proposed method.

**Weaknesses:**

1. Data access/movement energy consumption was not taken into account.
The energy consumption discussed in Fig. 3/5/6/7/8 are based on "computation cost", where the potential energy saving from FP32 to INT4 is estimated from reference Horowitz2014. But in the same reference it also shows that the cost for memory access is ~10-100x higher than mul-add computation. This missing piece is critical because the proposed method, as illustrated in Fig. 4, may need do either a on-the-fly quantization from FP32 weights to INT4 weights (which will require mem allocation) or hold 2 sets of weights (one for INT4 and one for FP32) for re-computation purpose. At least the model/kernel will need to do additional (GPU DRAM) access N time, where N equals to the number of elements being recomputed. Furthermore, for generative LLM it is very common that the inference is memory-bound instead of compute-bound, which means the memory access cost would very likely dominate the total energy consumption. This is exactly the reason why GPTQ inference can be done in FP16 but still achieve close to 4x speed-up, as the memory access cost is greatly reduced by employing INT4 weights.


2. Overhead of sparsity-like HW implementation.
The author suggests that the partial re-computation could be performed in a similar way to 2:4 structural sparsity. However, the author also acknowledges that this sparsity approach will require "mask tensors" of the same size as the input tensors to matmul engine, which will incur additional data movement cost. One should note that these two mask tensors related cost is adding on top of the data accessing cost mentioned above. This again highlights the importance of including data accessing cost in the energy consumption analysis.

**Questions:**

Please address the concerns mentioned in Weakness above.

---

### Meta-Review · Area_Chair_AZZw · 2024-12-18

**Metareview:**

The paper presents a promising approach to reducing the energy footprint of transformer models through nonlinearity-aware quantization. All reviewers have provided consistently negative ratings, regarding the practical implementation of NAQ, the accuracy of energy consumption estimates, the lack of consideration for data movement costs, the need for more explicit guidance on hyperparameter settings etc. The authors did not provide a response to address these issues. The final consensus of negative ratings lead to a rejection for this submission.

**Additional Comments On Reviewer Discussion:**

The authors did not provide a response to address the original concerns from reviewers. All reviewers remain the original ratings.

---

### Decision · Program_Chairs · 2025-01-22

Reject